# Spatial and Temporal Variability of Minimum Brightness Temperature at the 6.925 GHz Band of AMSR2 for the Arctic and Antarctic Oceans

Young-Joo Kwon [1] , Sungwook Hong [2] , Jeong-Won Park [1], Seung Hee Kim [1] , Jong-Min Kim [1] and Hyun-Cheol Kim [1,*]

[1] Center of Remote Sensing and GIS, Korea Polar Research Institute, Incheon 21990, Korea; kwonyj@kopri.re.kr (Y.-J.K.); jeong-won.park@kopri.re.kr (J.-W.P.); seunghee@kopri.re.kr (S.H.K.); poko519@kopri.re.kr (J.-M.K.)

[2] Department of Environment, Energy and Geoinfomatics, Sejong University, 209 Neungdongro, Gwangjin-gu, Seoul 05006, Korea; sesttiya@sejong.ac.kr

\* Correspondence: kimhc@kopri.re.kr

**Abstract:** The minimum brightness temperature (mBT) of seawater in the polar region is an important parameter in algorithms for determining sea ice concentration or snow depth. To estimate the mBT of seawater at 6.925 GHz for the Arctic and Antarctic Oceans and to find their physical characteristics, we collected brightness temperature and sea ice concentration data from the Advanced Microwave Scanning Radiometer 2 (AMSR2) for eight years from 2012 to 2020. The estimated mBT shows constant annual values, but we found a significant difference in the seasonal variability between the Arctic and Antarctic Oceans. We calculated the mBT with the radiative transfer model parameterized by sea surface temperature (SST), sea surface wind speed (SSW), and integrated water vapor (IWV) and compared them with our observations. The estimated mBT represents the modeled mBT emitted from seawater under conditions of 2–5 m/s SSW and SST below 0 °C, except in the Arctic summer. The exceptional summer mBT in the Arctic Ocean was related to unusually high SST. We found evidence of Arctic amplification in the seasonal variability of Arctic mBT.

**Keywords:** passive microwave; ocean; AMSR2; tie point; Arctic amplification





## 1. Introduction

The brightness temperature (BT) of seawater measured by passive microwave (PMW) satellite data is approximately proportional to sea surface emissivity and sea surface temperature. The emissivity of seawater depends on the salinity of the sea surface (SSS), sea surface temperature (SST), and sea surface wind speed (SSW) for the given observation frequency band and incidence angle of the sensor. The polar region is where the phase transition occurs between seawater and sea ice. Because solar radiation incidents in the Arctic and Antarctic oceans are low, the SST can drop to the freezing point. Open water with a salinity of 35 ‰ starts to freeze below the freezing point of −1.75 °C. If seawater were colder than the freezing point, it would freeze in thermal equilibrium. When seawater is frozen and sea ice is formed, emissivity in the microwave region increases from a value of 0.2–0.5 for the seawater to 0.9 for sea ice. Therefore, even if the sea ice has the same temperature as the ocean, the BT of sea ice measures higher than that of the ocean. Therefore, we suggest that the minimum BT (mBT) should be the coldest region near the freezing point of seawater in the polar oceans.

The mBT values in the Arctic and Antarctic oceans can be used as endpoint members of the BT distribution. The parameters related to the endpoint members of seawater and sea ice have been utilized for monitoring sea ice in the cryosphere. The most popular products including sea ice concentration (SIC) and snow depth come from PMW satellite observation data using significant emissivity contrast between the seawater and sea ice in

a selected microwave region. The endpoint members, so-called tie-points, for the seawater and sea ice are essential for retrieving products related to sea ice and to snow over the sea ice in the Arctic and Antarctic regions. For example, the bootstrap algorithm [1] and NASA team algorithm [2] define the reference points of BT for 0% and 100% SIC using higher frequency bands. Ice concentrations of 0% and 100% correspond to the tie-points of open water and sea ice in the set of BT values, respectively. For the snow depth retrieval algorithm [3], the tie-points for the ocean at 18 and 37 GHz are necessary for obtaining accurate snow depth data. Therefore, the determination of tie-points for the ocean and sea ice from observed BTs are important for maintaining the accuracy and consistency in operational sea ice products. Despite broad usage of the tie-point concept, the physical interpretation of tie-points is somewhat unclear.

Generally, SIC products provide a good accuracy (precision) of less than 10%. However, there is high uncertainty in the retrieval products for the summer period because of surface melting and the presence of melt ponds. Atmospheric contributions and wind roughening of the open ocean are also significant error sources [4]. Thin ice in the marginal zone is another challenge for PMW algorithms [5]. Recently, products using the lower frequency bands in a PMW satellite have been proposed as a new alternative. Lower frequency bands are more noticeable for the detection of thin sea ice and less sensitive to atmospheric contribution and surface roughness than conventional higher frequency bands. Kilic et al. [6] proposed a new sea ice concentration algorithm, including observation data at 6.9 and 10 GHz bands, to improve SIC estimation. They showed that the algorithm using 6.9 GHz observations had the lowest rate of error [6,7]. A recent study by Rostosky et al. [8] showed improved accuracy for the output of a snow depth algorithm including the C-band. Up to now, the low spatial resolution of low-frequency bands (<11 GHz) has been seen as a drawback. However, ESA's planned Copernicus Imaging Microwave Radiometer (CIMR) mission is expected to include L- and C-band channels with improved resolution [9]. The higher spatial resolution at the lower frequency channels enables retrievals of various surface parameters with a lower level of uncertainty than the current operational retrieval products. Furthermore, it can maintain a sustained continuity of PMW measurements, which is required in the climate study.

The Arctic has been warming roughly twice as fast as the rest of Earth over the past 30 years [10]. The main drivers of the "Arctic amplification" [11] are the snow/ice-albedo and cloud-radiation feedbacks. Over the last 30 years, the sea ice in the Arctic Ocean has been shrinking [12], and land snow cover has decreased especially during the summer. The reduction in both sea ice and land ice in the Arctic has led to increases in SST [12] and atmospheric water vapor [13–15] as ice-free regions expand. Understanding the patterns of rapid changes in the polar environment detected by PMW sensors can provide a new methodology for monitoring climate change. In addition, it is possible to infer the effect of the changed physical variables in the polar region on the accuracy of the current retrieval products.

In this paper, we estimated the mBT and examined seasonal variability and regional differences between the Arctic and Antarctic Oceans on a global scale. Although the mBT uses the same concept as tie-points in the sea ice concentration algorithm, we use the mBT in this paper instead of tie-points. The tie-points generally refer to endmembers at 19 and 37 GHz for the sea ice concentration retrieval algorithm (e.g., bootstrap algorithm [1]). However, in this paper, we used BT at 6.925 GHz to analyze the effects of the physical properties. We collected the BT at the 6.925 GHz band measured by Advanced Microwave Scanning Radiometer 2 (AMSR2) onboard the Global Change Observation Mission–Water 1 (GCOM-W1) satellite for eight years from 2012 to 2020. For comparison, the theoretical seawater emissivity and BT were calculated using the dielectric constant model of Meissner and Wentz [16].

## 2. Materials and Methods

AMSR2, the successor of the AMSR-E onboard the Aqua satellite, is the PMW radiometer system onboard the Japan Aerospace Exploration Agency's GCOM-W1 satellite. The GCOM-W1 was launched in 2012 [17] and has been orbiting in a sun-synchronous low-Earth orbit with an inclination angle of 98.2°. AMSR2 observations are performed by conical scanning with a constant incidence angle of 55° and a swath width of 1450 km. The AMSR2 has seven microwave frequency bands at 6.925, 7.3, 10.65, 18.7, 23.8, 36.5, and 89.0 GHz in the H- and V-polarizations [18]. While the nadir footprint size at the 6.925 GHz channel of the AMSR2 is coarse ($35 \times 62$ km$^2$), it is sufficient for interpreting polar observation data because it is less affected by the atmosphere and has a relatively high emissivity contrast between the sea ice and seawater.

We collected the brightness temperature (Level 1B) and the Sea Ice Concentration products (Level 2 [19]) from the AMSR2 at 6.925 GHz for eight years from July 2012 to June 2020 (see https://gportal.jaxa.jp/, (accessed on 27 May 2021)), except for three days of missing data (2013.05.11–13). We used both observation data in ascending and descending orbits without distinction between them. We excluded sea ice (SIC > 0), land, and the contaminated data adjacent to the land or sea ice from the mBT estimate. Finally, data from seawater at high latitudes (above 55° N for the Arctic Ocean and 45° S for the Antarctic Ocean) were gathered separately from the daily collected data. The estimation of the mBTs for V- and H-polarizations were determined by averaging the lower 0.1% of data to reduce potential noise in daily values. The lower 0.1% of the data set was sorted by projecting the data points onto the trend-line of the BT in the domain of the BTs at V- and H-polarizations. We conducted an orthogonal projection onto a regression line. Assuming BTs for V- and H-polarization as the position vector ($V$, $H$) in the $BT_V$ and $BT_H$ space, the orthogonal projection onto the regression line can be described as follows:

$$\begin{pmatrix} U_t \\ U_s \end{pmatrix} = \begin{pmatrix} \sqrt{1/(1+s)^2} & \sqrt{s^2/(1+s)^2} \\ -\sqrt{s^2/(1+s)^2} & \sqrt{1/(1+s)^2} \end{pmatrix} \begin{pmatrix} V \\ H \end{pmatrix} \tag{1}$$

where $s$ indicates the slope of the regression line. $U_t$ and $U_s$ are the position vectors in a new space defined by the unit vectors parallel or perpendicular to the linear regression line. $U_t$ indicates the projected position on the regression line, and the average of the lower 0.1% based on $U_t$ was determined as mBT. At least 1000 data points were available to get the lower 0.1% dataset in the transformed domain. Figure 1 shows an example scatterplot for BT in each Arctic (left) and Antarctic (right) pole on 3 November 2018. The grey and the cyan circles in Figure 1 represent the BT data points and the computed mBT, respectively. The dashed line in Figure 1 indicates the trend line of BT data points in the polar ocean obtained by the linear regression method.

To examine the factors inducing seasonal variability, The Remote Sensing Systems (RSS) model [16,20] was used for the framework of RTM simulation in this paper. It is the representative ocean RTM developed with SSMI and WindSat observation [21]. We applied the wind-induced sea surface emissivity model [16] but did not consider the effect of seafoam and sea surface wind direction. According to the RTM, BT measured by passive microwave sensors includes radiation energy emitted from various sources. For a non-scattering plane-parallel atmosphere, BT observed from the top of the atmosphere from the ocean surface consists of three radiation components: (1) surface-emitted radiation attenuated by the atmosphere, (2) upwelling and surface-reflected downwelling atmospheric radiation, and (3) surface-reflected cosmic background radiation. The radiative transfer equation in the PMW from the ocean surface can be written as

$$BT_p = e_p^R \Gamma T_s + T_u + \left(1 - e_p^R\right)\Gamma T_d + \left(1 - e_p^R\right)\Gamma^2 T_{CMB} \tag{2}$$

where the subscript p denotes the vertical (V) or horizontal (H) polarization component. $T_s$ is the SST, and $e_p^R$ is the rough surface emissivity of the ocean. $T_{CMB}$ is the BT of the

cosmic background from space (approximately 2.7 K). $\Gamma$ is the atmospheric transmittance defined as $\Gamma = \exp(-\tau_a/cos\theta_i)$, where $\tau_a$ is the optical depth of the atmosphere and $\theta_i$ is the Earth incident angle of the PMW sensor. The atmospheric optical depth ($\tau_a$) along the atmospheric slant path depends on the optical depth of oxygen, water vapor profile, and cloud liquid water [22,23]. $T_u$ and $T_d$ are the temperatures of the atmosphere responding to upwelling and downwelling radiation components, respectively. For the PMW sensor, $T_u$ and $T_d$ can be approximated by the atmospheric effective temperature ($T_a$), which strongly relies on the vertical distribution of atmospheric parameters such as atmospheric temperature, relative humidity, and liquid water contents [22,24].

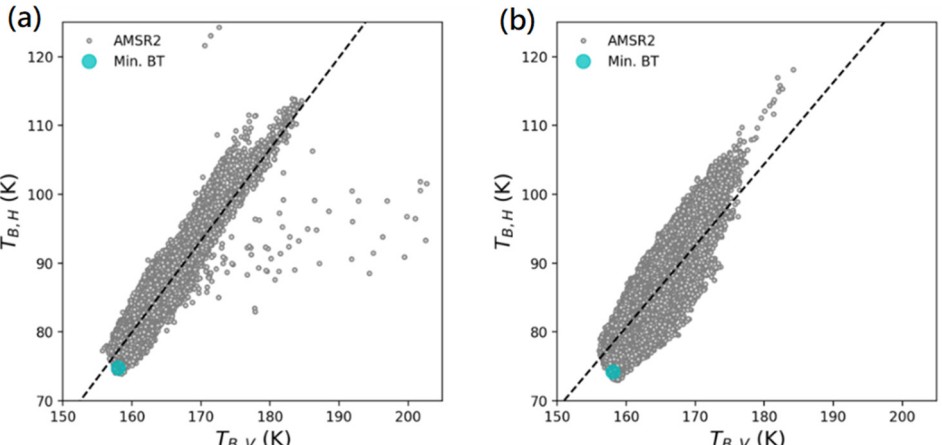

**Figure 1.** Scatterplots of brightness temperature (BT) and the minim BT for the Arctic (**a**) and Antarctic Ocean (**b**).

The most dominant component is radiative energy from the surface of the ocean. The surface term in (2) can be expressed as reflectance for a given polarization.

$$e_p^R \Gamma T_s = \left( e_p^S + \Delta e_W \right) \Gamma T_s \tag{3}$$

The SST in the polar regions ranges from the freezing point (about $-1.75\,°C$ in 35‰ seawater) to 10 °C. $e_p^R$ indicates the emissivity of the rough ocean surface. A notable approach to the surface height distribution of rough surfaces is described by a Gaussian random distribution with zero mean and $\sigma_{rms}$ standard deviation [25–27]. According to this description, the reflectivity on the rough surface ($R_p^R$) is related to a specular surface reflectivity ($R_p^S$) and roughness parameter, which is a function of the root mean square height ($\sigma_{rms}$), the wavelength, and the incidence angle ($\theta_i$). As the root mean square height increases, rough surface reflectance becomes lower but emissivity increases. The specular surface emissivity on the ocean surface, $e_p^S = \left( 1 - R_p^S \right)$, is governed by the Fresnel formula for a certain complex dielectric constant at a local incident angle.

The specular surface emissivity of the ocean is the most dominant component, which the Fresnel formulas describe as emission. As the SSW blows over the ocean's surface, small-scale capillary waves form and surface roughness increases, which leads to a dramatic change in the emission characteristics as a function of the SSW [28–31]. For ocean surface emissivity, there are three different types of roughness scales influencing surface roughness: (1) large-scale roughness due to gravity waves, (2) small-scale roughness due to gravity-capillary waves or SSW, and (3) seafoam. Large-scale gravity waves are important for observation when the wavelength of the ocean waves is longer than the wavelength of the incident radiation. These large-scale waves can mix vertical and horizontal polarizations and change the local incidence angle of the radiation. The small-scale gravity–capillary waves scatter incident radiation over the large-scale gravity waves at the surface of the ocean. This also affects the emissivity of the sea surface, which is treated as a perturbative

parameter at the surface of the ocean. Seafoam consisting of air bubbles and water can lead to an increase in surface emissivity. This effect is dominant under strong surface wind (above 7 m/s). Because we focus on the mBT of the Arctic and Antarctic Ocean in this paper, we do not consider the effects of the strong wind and seafoam on the ocean surface or precipitations responsible for increasing emission. We discuss wind-induced roughness at the centimeter length scale (capillary waves) at the 6.925 GHz band but ignore the effect of wind direction variation.

In this study, we used the SSW-induced model of Meissner and Wentz [16] to determine the effect of seawater emissivity change via SSW. Meissner and Wentz presented emissivity models for the wind-roughened ocean surface for microwave frequencies between 6 and 90 GHz. They used TB measurements from the space-borne microwave radiometer WindSat and the Special Sensor Microwave/Imager (SSM/I) as the basis of analysis. They provided an empirical isotropic wind-induced emissivity model with a fifth-order polynomial form ($\Delta e_W = \sum_{k=1}^{5} \delta_k SSW^k$). Because the SSM/I has an incidence angle and bands that are similar to AMSR2, we used the isotropic wind-induced emissivity model without any modification.

To estimate the specular reflectivity of the ocean surface, the permittivity of seawater is necessary. The permittivity of seawater is generally assumed to be of a dielectric medium described by the Debye relaxation model. The Debye relaxation model can accurately describe the permittivity of dielectric material, which is a function of the external electromagnetic wave frequency and relaxation time ($\tau$) of the material. The dielectric constant model of seawater has been developed and significantly improved by many previous studies and the Debye relaxation model [32–34]. The parameters in the Debye relaxation model ($\varepsilon_s$, $\varepsilon_\infty$, and $\tau$) depend on the temperature and the salinity of the ionic salts. We used coefficients for the dielectric constant model with double Debye relaxation wavelengths by Meissner and Wentz [20]. The dielectric model requires inputs of the microwave frequency, SST, and SSS to calculate the complex dielectric constant.

Under clear sky conditions at microwave frequency bands lower than 10 GHz, atmospheric contributions to the BT in satellite observations are relatively small [35–37]. However, the atmospheric influence can induce seasonal variability because the amount of water vapor and cloud liquid water in the atmosphere over the ocean is usually large during the summer and early fall. The atmospheric parameters ($\tau_a$ and $T_a$) in (2) can be calculated from the given atmospheric profiles of temperature and the absorption coefficients of each component. The atmospheric optical depth ($\tau_a$) can be accurately parameterized as a function of oxygen optical depth, water vapor mass absorption and liquid water mass absorption coefficients [38,39] as follows:

$$\tau_a = A_{O2} + A_{WV} + A_{LW} = \int_0^H (\alpha_{O2} + \alpha_{WV} + \alpha_{LW})dh \tag{4}$$

where $A_{O2}$, $A_{WV}$, and $A_{LW}$ indicate contributions to optical depth by Oxygen ($O_2$), water vapor (WV), and liquid water (LW) in the atmosphere, respectively. The $\alpha_i$ denotes the absorption coefficients for the $i$ component of $O_2$, WV, or LW. Although the parameters are related to atmospheric profiles, we use the approximation relationships of Meissner [40] from 1-dimensional RTM results for horizontally uniform atmospheric profiles of temperature and humidity, which depend only on the altitude above the surface. The approximations for $T_a$, vertically integrated oxygen absorption ($A_{O2}$) and the vapor absorption ($A_{WV}$) are parameterized by the vertically integrated water vapor (IWV) in mm and SST.

To calculate the atmospheric influence in RTM simulations, the atmospheric component was assumed to be a clear sky without clouds or rain. Under cloudless clear sky conditions, the liquid water absorption term becomes zero. The assumption minimizes the contribution of liquid water absorption ($\alpha_{LW}$) in the atmospheric term—even if a small amount of liquid water is applied to the model, it calculates BT easily exceeding mBT, which is not relevant to this paper.

SST, SSW, and IWV were selected as input parameters for the RTM simulation. The sensitivity of the geophysical parameters to the BTs at 6.925 GHz varies with frequency and polarization. Generally, the sensitivity of SST and SSW are large for the V- and H-pol. The SSW affects the emissivity of the sea surface by inducing sea surface roughness [16,30]. The SSW has larger sensitivities to H- than V-polarization, and there is a distinct change in slopes for V-polarization at about 7 m/s. The sensitivity of the BT at C-band to SST is larger at V-polarization than H-polarization [41]. The influence of water vapor change is smaller than that of SST or SSW at low frequencies. The SSS and relative SSW direction have a minor effect on the BT. In this paper, we assumed the SSS to be 34‰ in both polar regions.

## 3. Results

Overall, the mBT showed a stable value without significant temporal/spatial variability. The estimated mBTs in the Arctic Ocean were 157.852 K ($\pm$0.485 K) for V-polarization and 74.949 K ($\pm$0.612 K) for H-polarization. For the Antarctic Ocean, the results were similar but slightly lower than those in the Arctic Ocean: 157.654 K ($\pm$0.330 K) for V-polarization and 74.784 K ($\pm$0.748 K) for H-polarization. These small standard deviations imply a stable feature for the extended period of 8 years from July 2012 to June 2020. Figure 2 shows the boxplots of the annual mBT for the V- and H-polarization in the Antarctic and the Arctic Ocean. The horizontal gray-colored lines in Figure 2 indicate the mean mBT for each case. The cyan and orange color circles indicate outliers. The difference in mBTs between the Arctic and Antarctic Oceans seems subtle, but statistically, the two samples show a significant difference at a 95% confidence level ($p$-value = $2.319 \times 10^{-49}$ for V-pol., $6.214 \times 10^{-11}$ for H-pol.). Since neither of the V-polarized and H-polarized data of the Arctic and the Antarctic Oceans passed the normality test, we conducted a non-parametric Wilcoxon signed-rank test [42] on two independent mBT samples for the Arctic and the Antarctic Oceans.

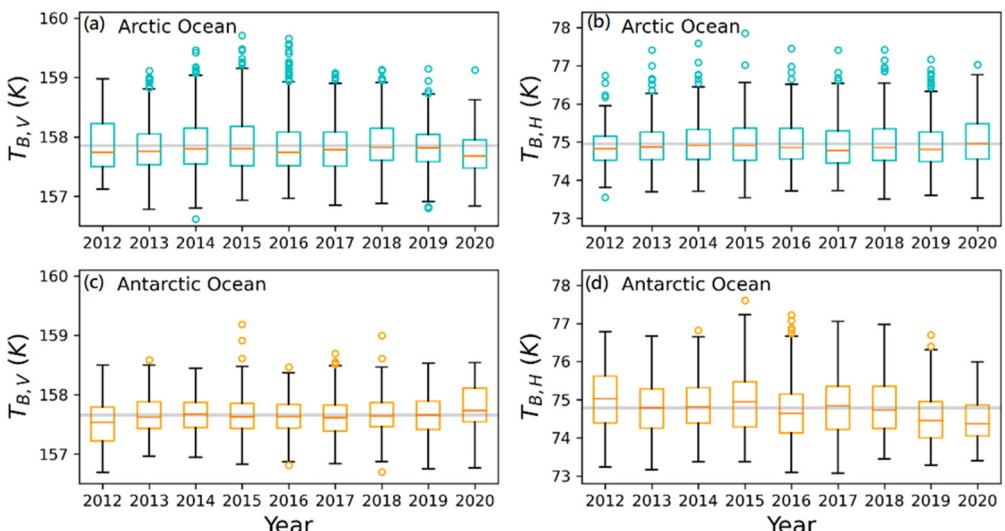

**Figure 2.** Boxplots for Annual mBT for the Arctic ((**a**) for V-polarization and (**b**) for H-polarization) and Antarctic Oceans ((**c**) for V-polarization and (**d**) for H-polarization).

Table 1 lists the monthly averaged mBT in the polar region for eight years. The mBT is comparable to the seawater tie-points from the previous study [7]. The sensors have different characteristics and slightly different incidence angles for the different observation periods. Nevertheless, the tie-points at 6.9 GHz of the AMSR-E and SMMR by Ivanova et al. [7] are similar to our results (Table 1). In this paper, the AMSR2 mBT of the V-polarization was estimated to be higher than the tie-points of the AMSR-E and SMMR, but lower for the H-polarization. The mBT difference between the Antarctic and the Arctic

Oceans was smaller than that of Ivanova's tie-point. Because the tie-points are affected by the seasonal variation and sensors and environmental change due to climatic trends in the surface and atmosphere, the dynamic tie points are usually used for the SIC algorithm. Ivanova et al. [7]'s tie-points in Table 1 developed a particular set of static tie-points to perform a fair comparison of the SIC algorithms.

**Table 1.** The monthly averaged mBT in Kelvin for eight years in the Arctic and Antarctic Oceans.

| Month | Arctic Ocean | | Antarctic Ocean | |
|---|---|---|---|---|
| | V-pol. (K) | H-pol. (K) | V-pol. (K) | H-pol. (K) |
| 1 | 157.685 (0.298) | 75.393 (0.572) | 158.147 (0.169) | 73.919 (0.312) |
| 2 | 157.617 (0.315) | 75.446 (0.543) | 157.904 (0.226) | 74.174 (0.367) |
| 3 | 157.595 (0.302) | 75.016 (0.561) | 157.652 (0.232) | 74.541 (0.453) |
| 4 | 157.523 (0.300) | 74.659 (0.405) | 157.492 (0.192) | 74.886 (0.540) |
| 5 | 157.706 (0.331) | 74.518 (0.371) | 157.473 (0.215) | 74.806 (0.533) |
| 6 | 158.154 (0.289) | 74.575 (0.412) | 157.482 (0.258) | 74.983 (0.627) |
| 7 | 158.639 (0.345) | 74.711 (0.407) | 157.520 (0.256) | 75.245 (0.695) |
| 8 | 158.404 (0.388) | 74.722 (0.402) | 157.540 (0.365) | 75.320 (0.610) |
| 9 | 157.845 (0.253) | 74.569 (0.384) | 157.518 (0.299) | 75.373 (0.747) |
| 10 | 157.637 (0.203) | 74.862 (0.468) | 157.522 (0.287) | 75.303 (0.593) |
| 11 | 157.719 (0.279) | 75.378 (0.551) | 157.671 (0.258) | 74.801 (0.667) |
| 12 | 157.676 (0.314) | 75.560 (0.654) | 157.933 (0.202) | 74.040 (0.433) |
| Total | 157.852 (0.458) | 74.949 (0.612) | 157.654 (0.330) | 74.784 (0.748) |
| AMSR-E [1] | 161.35 | 82.13 | 159.69 | 80.15 |
| SMMR [1] | 153.79 | 86.49 | 148.60 | 83.47 |

[1] Data from Ivanova et al. [7].

Figure 3 shows the seasonal variability of mBT in boxplots for the monthly mBT samples of the Arctic (upper panel) and Antarctic Ocean (lower panel). The horizontal gray lines in each plot of Figure 3 are the average mBT values for the entire study period. The circles in Figure 3 indicate outliers. Note that the Antarctic boxplot starts in July to match the seasonal changes between the Arctic and the Antarctic Oceans. In Figure 3, mBT of V-polarization increases in summer in both Antarctica and the Arctic, but H-polarization tends to decrease slightly. The mBT of H-polarization for the Antarctic Ocean shows a minimum in summer (December or January); however, it remains at a similar level from April to September for the Arctic Ocean.

We illustrate seasonal mBT variation in the Arctic and Antarctic Oceans and the result of RTM simulations in Figure 4. The cyan- and orange-colored symbols in each plot represent the mBTs of the Arctic and Antarctic Oceans, respectively. The vertical solid curves indicate BT with the same SST, corresponding to −1.8, −1, −0.2, 0.6, and 1.4 °C from left (blue) to right (red). The horizontal dashed lines show the simulated BT with the same SSW corresponding to 0, 1, 2, 3, 4, and 5 m/s from bottom to top. The atmospheric conditions in RTM simulations were applied differently for each season: IWV = 5 mm for spring and autumn, 3 mm for winter, and 12 mm for summer. Overall, the SST of the estimated monthly mBTs remained below 0 °C for all seasons except summer (July and August) in the Arctic Ocean. The monthly mBT in August reached a relatively high SST of up to 0.6 °C. The Antarctic Ocean's monthly mBTs maintain a low SST near the −1.0 °C curve. SSW in both the Arctic and Antarctic Oceans showed a pattern of weakening in the summer and strengthened in the winter. In the summer, a weak SSW of less than 2 m/s is dominant in both oceans. The SSW tends to increase from autumn (less than 2 m/s in Figure 4c) to winter (up to 3 m/s in Figure 4d) but decreases from winter to spring (about 2 m/s in Figure 4a).

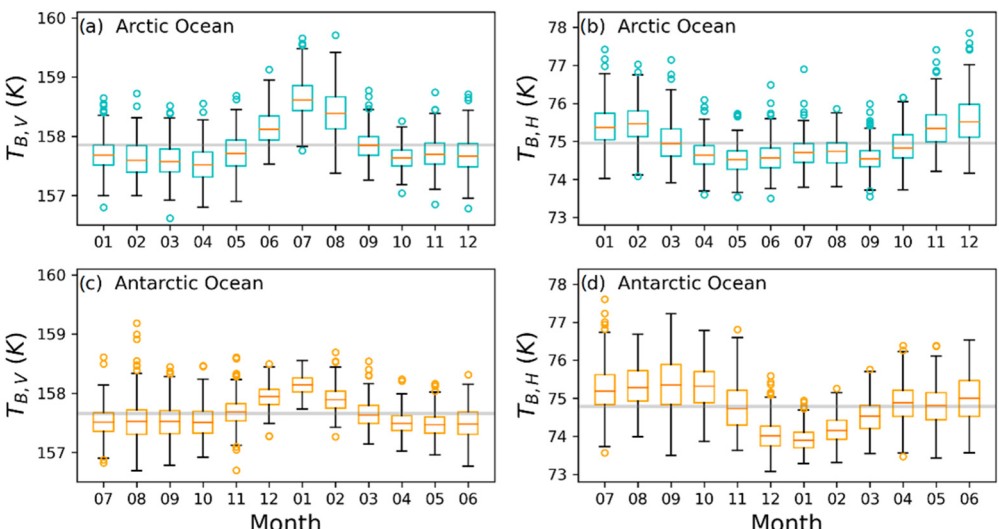

**Figure 3.** Boxplots for Monthly mBT for the Arctic ((**a**) for V-polarization and (**b**) for H-polarization) and Antarctic Oceans ((**c**) for V-polarization and (**d**) for H-polarization).

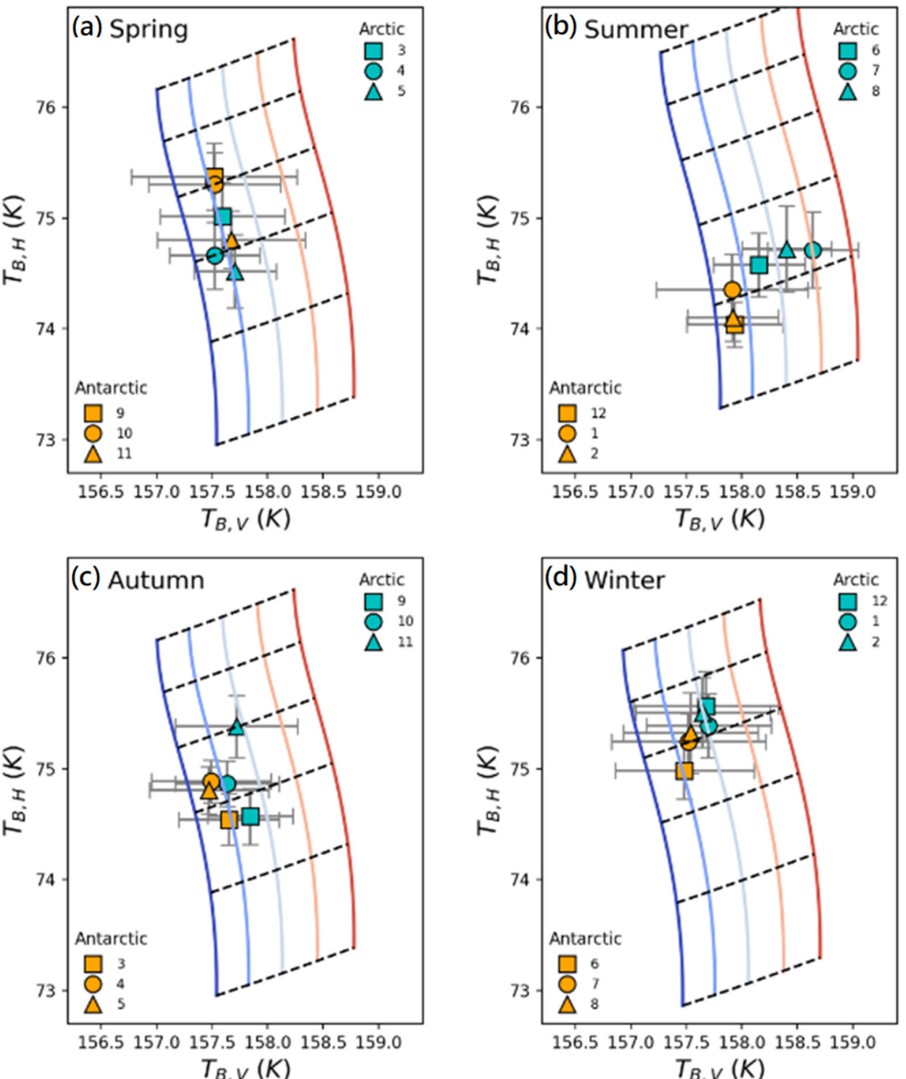

**Figure 4.** The seasonal variation of mBT and calculated BT via RTM simulations for comparison.

We tried estimating monthly SST and SSW for given IWV as follows. The IWV was obtained from the lower 10th percentile of ECMWF reanalysis data set (ERA5) for eight years from 2012 to 2020. Then, we can estimate the data set (SST, SSW, IWV) that best describes the estimated mBT of V- and H-pol. Figure 5 shows the monthly estimated SST and SSW by comparison of RTM simulations with monthly mBTs, and IWV as input data. We illustrated the SST and SSW from ERA5 (yellow squares for lower fifth percentile and green squares for 10th percentile) for comparison. The retrieved SST and SSW for the Antarctic Ocean were similar to the ERA5 percentiles, but those for the Arctic Ocean did not agree. The difference between the two polar oceans seems to be due to the different frequency distributions for each variable.

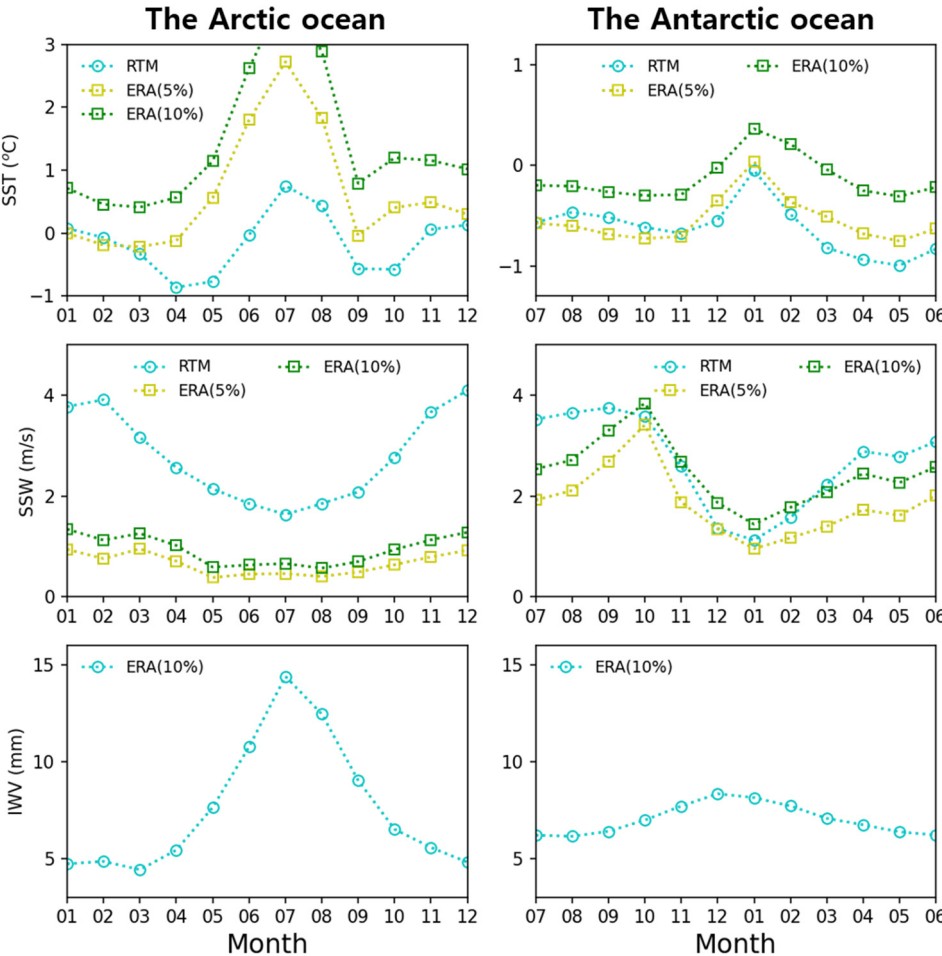

**Figure 5.** The estimated monthly SST and SSW by comparison RTM simulations with monthly mBTs and IWV as input data.

## 4. Discussion

From the BT data measured by AMSR2 for eight years, we found a similar mBT in the Arctic and Antarctic Oceans: 157.852 K (Arctic) and 157.654 K (Antarctic) for V-polarization and+ 74.949 K (Arctic) and 74.784 K (Antarctic) for H-polarization. While the mBT differences between the Arctic and Antarctic Oceans were about 0.2 K in both polarizations, the differences were statistically significant at the 95% confidence level. Our estimate coincides with the previous work of Ivanova et al. [7], which showed that all tie-points for the Antarctic Ocean from SSMI and AMSR-E data are lower than that of the Arctic Ocean.

The main achievement of our work is that we identified SST as a main controlling factor causing the difference in mBT between the two regions, which is shown by comparing

the observed mBT with that of RTM simulation. Our finding is also supported by the previous work of Carvalho and Wang [43], in which the relatively higher seasonal mean Arctic Ocean SSTs of summer (2.82 °C) and autumn (2.15 °C) are reported than those of the winter (0.12 °C) and Spring (0.11 °C). The spatial pattern of seasonal SST is characterized by the relatively high SST in the Atlantic side of the Arctic (e.g., the Norwegian Sea and Greenland) and low SST in the Pacific side of the Arctic Ocean. In the summer, the Barents and the Norwegian Seas show a high SST of 4–11 °C [43].

In contrast to the Arctic oceans, SST in the Antarctic Ocean maintains temperatures as low as 0 °C even in summer. According to Shao and Ke [44], the highest SST record from spring to summer (Nov. to the following Feb.) is −0.8 °C in the Indian Ocean sector. The Antarctic Ocean's warming trend is also not apparent. Comiso et al. [45] estimated the overall trend of 0.1 °C per decade for the surface temperature in the entire Southern Ocean and an increasing trend (1.73% per decade) for sea ice extent in the Antarctic Ocean. However, Lebedev [46] reported a negative trend of SST (–0.2 °C/decade) over the entire Southern Ocean.

Another important factor affecting the position of the mBT is the SSW. Even if the sea surface has a low temperature near the freezing point, the BT can be larger if a strong wind blows over the ocean surface. Both conditions of low SST and weak SSW are necessary to obtain the mBT. The dependence of mBT on the SSW weakens in the summer and intensifies in the winter. In general, the Arctic Ocean has numerous robust cyclones in the winter but weaker and less frequent ones in the summer (e.g., [47–49]). Arctic cyclones are most dynamically intense during the winter [50]. The seasonal variability of the SSW in this paper agrees with the general SSW pattern.

The Antarctic Ocean also shows similar seasonal variations in wind speed patterns (e.g., [51–53]). The number of occurrences and the magnitude of the variability of the SSW are much smaller in the summer compared to the other seasons [51]. Laurila et al. [54] found seasonal variation with the strongest winds during the winter months at a monthly mean of 10 m wind speed over 40 years (1979–2018). They reported that the monthly distribution of 10 m wind speed in the central North Atlantic shows seasonal variability with the highest winds during winter and the lowest in the summer.

Sea ice cover plays an essential role in energy exchanges between the atmosphere and the ocean in the cryosphere. The white ice surface with a high albedo reflects far more sunlight into space than ocean water does. Sea ice also creates an insulating cap across the ocean surface, which reduces evaporation and heat loss from the ocean into the atmosphere. The sea ice has been interacting with the other components of the climate system. As a result, the weather over ice-covered ocean regions tends to be colder and drier than over ocean regions without sea ice.

However, Arctic sea ice has been declining at a rate of ~3.8% per decade [55]. The declining trends in sea ice cover and thickness [56] coincide with Arctic amplification. Many others have reported the warming trend in the Arctic Ocean. Comiso [57] found a positive trend at 0.33 °C per decade over the Arctic sea ice from the satellite observation for 20 years from 1981 to 2001. Carvalho and Wang [43] reported a similar trend of 0.36 °C per decade using the data from 1982 to 2018. The surface temperature including land in the Arctic region increased more rapidly at 0.60 °C per decade.

The change has been most dramatic in the summer and autumn periods [58]. The warming of the Arctic Ocean in the summer is caused by the heating of surface water by seasonal cycle solar radiation in ice-free regions [59] and improved vertical ocean heat transport [60]. The Arctic Ocean also receives warm inflows from the rivers and the Atlantic and Pacific Oceans (e.g., [61,62]). The atmospheric response to reducing sea ice cover is related to low-level cloud formation [63] and increasing levels of liquid water vapor [64]. Evaporation over ice-free ocean regions leads to an increase in atmospheric water vapor and low-level clouds (e.g., [13–15,65]). For these reasons, relating to intensified atmospheric water vapor and clouds, together with higher SST, this effect contributes to the exceptional mBT value of the Arctic summer season.

The seawater tie-point represents a typical signature of the sea surface on a hemispheric scale. Deviations from the specific surface properties result in ice concentration uncertainties. The dynamical tie-point based on seawater's actual mean signature has been applied to reduce spatial and temporal variability. Moreover, the dynamic tie-points tune dynamically for the different instruments and simplify the implementation of other sensors. However, our results show that the mBT in the Arctic summer departs from the typical seawater signature for the sea ice concentration algorithm. It will further depart from mBT in the summer as SST and IWV increase, along with reductions in Arctic sea ice. The seawater tie-point in the summer may become somewhat higher in the current algorithm due to the dynamical tie-point method. The accuracy of SIC in the summer season will be worse and may gradually expand to other seasons.

The mBT depends on essential climate variables such as the SST, SSW, and IWV in the polar ocean. The characteristic distribution of the mBT in the Arctic summer seems to be related to Arctic amplification consequences. The fact that drastic changes in the polar climate are captured in the PMW satellite-based mBT suggests a potential tool for comprehensively monitoring polar climate change. Furthermore, the BT's estimate according to the PMW satellite has excellent advantages in terms of processing, being faster and more accurate than using retrieval products in a level 2. We expect that the results in this paper will lead to a better understanding of Arctic routes and climate change.

The 18 and 36 GHz bands used in the SIC algorithm are more sensitive to tie point changes than the 6.925 GHz band. According to Kilic et al. [6], the point will directly impact both the systematic and random errors of the SIC retrieval. They showed that changes in the tie points could induce biases in the SIC estimate up to ~8% with related standard deviations up to 9% for retrieval algorithm using 18 and 36 GHz bands. The Arctic sea ice reduction trend could involve higher SST and wetter and more cloudy atmospheric conditions in the future Arctic Ocean. The changed Arctic Ocean environment will lead to higher summer tie points and, consequently, negatively affect the accuracy of the SIC estimation.

## 5. Conclusions

We collected data of GCOM-W1/AMSR2 BT at 6.925 GHz and SIC data for eight years from July 2012 to June 2020. We determined the mBT for V- and H-polarization with the average of the lower 0.1% of data based on the points projected along the trend line. We found a similar mBT in the Arctic and Antarctic Oceans: 157.852 K (V-pol.) and 74.949 K (H-pol.) for the Arctic and 157.654 K (V-pol.) and 74.784 K (H-pol.) for the Antarctic. The mBTs are stable annually over the study period, but there are statistically significant differences in mBT between the Arctic and Antarctic Oceans at the 95% confidence level. We also found subtle seasonal variability of about 1 K in the mBT for each region. In the Arctic and Antarctic Ocean regions, there is a tendency for an increasing V-polarization mBT but a decreasing H-polarization mBT during the summer season. The only difference in seasonal variability trends between the Arctic and Antarctic Oceans is that the mBT of the H-polarization did not decrease as much as that of the Antarctic ocean.

We estimated the mBT at 6.925 GHz using the RTM simulations to discuss the cause of the seasonal variability. The coefficients for the dielectric constant model by Meissner and Wentz [20] and the Meissner and Wentz [16] wind-indued roughness surface models were used to calculate the BT. We parameterized SST, SSW, and IWV as input variables to compute RTM simulations. Via comparison with the RTM simulations, we found that the SST in the polar region is constrained below 0 °C except for during the summer (July and August). The RTM simulation results can explain the slight seasonal variability in mBT by influencing SST, SSW, and IWV variation. Relatively high SST and IWV but weakening SSW leads to mBT characteristics of departure from the common position in the summer season of the Arctic Ocean.

Over the last few decades, the Arctic Ocean region has experienced extreme climate change. As the climate situation in the polar regions changes dramatically, the influence of

the polar regions is extending to the weather in the mid-latitudes and the global radiative energy budget. The minor spatial-temporal difference we found could be related to climate change. Our results provide new evidence for rapid climate change in the Arctic Ocean.

Our findings can be used to improve the accuracy of the SIC and snow depth retrieval algorithm and lead to the development of the C-band-based SIC and snow depth algorithm. C-band observation data by a PMW sensor are sensitive to sea ice while having a weak atmospheric effect. Since it has been observed for a relatively longer time than the L-band observation, it is expected that it can be used for long-term variability studies, and the L-band is sensitive to SST change and the sea ice thickness whereas 6.9 GHz is not. Radio Frequency Interference is mitigated when using low frequency such as L- and C-band channels. Furthermore, both bands are insensitive to atmospheric conditions. L/C-band combination will provide a considerable synergy to solve uncovered geophysical variables.

**Author Contributions:** Conceptualization, Y.-J.K.; methodology, S.H. and J.-W.P.; investigation, S.H., J.-W.P., S.H.K., J.-M.K., and H.-C.K.; resources, Y.-J.K.; writing—original draft preparation, Y.-J.K.; writing—review and editing, S.H., J.-W.P., S.H.K., J.-M.K., and H.-C.K.; visualization, Y.-J.K.; project administration, H.-C.K.; funding acquisition, H.-C.K. All authors have read and agreed to the published version of the manuscript.

**Funding:** This research was supported by the Korea Polar Research Institute (KOPRI) through grant PE21040 and the Ministry of Trade, Industry and Energy (MOTIE, Korea) (Project Number: PN21030).

**Institutional Review Board Statement:** Not applicable.

**Informed Consent Statement:** Not applicable.

**Data Availability Statement:** The data and code used in this study are available by contacting the corresponding author.

**Conflicts of Interest:** The authors declare no conflict of interest.

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
