# Peer review of "Spatial and Temporal Variability of Minimum Brightness Temperature at the 6.925 GHz Band of AMSR2 for the Arctic and Antarctic Oceans"

_remotesensing, doi:10.3390/rs13112122_

Round 1
Reviewer 1 Report
This manuscript presents the estimation results of polar ocean (Arctic & Antarctic) mBT at 6.9GHz from AMSR2 observation from 2012 to 2020. The manuscript also discusses several further extending issues, including the dependence of mBT on SST & SSW, the difference between Arctic & Antarctic oceans, and the seasonal variation feature. The topic of this paper is interesting, since ocean mBT (tie-point) plays a very import role in SIC retrieving and it’s long term variation may reflect the “arctic amplification” effects under the scenario of the accelerated global warming. The main weakness of the paper is its inadequate scope and depth. The discussion and conclusion are constrained to BT (brightness temperature), not extended to any real physical environmental variable (SIC for example). However BT is much related to a specific instrument, not suitable to be used as an indicator for climate change monitoring. The significance of the paper will be much improved if the validation can be done on SIC.Author Response
Dear Reviewer
We would like to thank you for the kind review. We are really grateful for the comments and excellent suggestions we have received and these made a much better paper. According to the comments, we have revised our manuscript. Please see the attachment.
Sincerely,
Hyun-Cheol Kim, Ph.D
Director, Center of RS & GIS, KOPRI

Reviewer 2 Report
General comments
This is a interesting study that describes the variability of the minimum brightness temperatures (mBT) of the ocean surface near sea ice of both Arctic and Antarctic Oceans. The study provides inputs for the understanding of the annual and seasonal variability of the ocean BT, and the results are well discussed. It gives a physical interpretation of the variability of the ocean tie point which is mainly used in sea ice concentration algorithms.
Specific comments
-Page 1, Introduction, line 4: The emissivity of seawater depends also on the sea surface wind speed and the incidence angle of the sensor.
-Page 1, Introduction, line 9: “When seawater is frozen...” This sentence is unclear for me, please reformulate maybe by: “When seawater is frozen and sea ice is formed, emissivity in the microwave region increases from a value of 0.2–0.5 for the seawater to 0.9 for sea ice.”
-Page 2, Introduction, last paragraph, line 4: “The tie-points in the SIC algorithm are corrected for the
influence of water vapor ...” It is not always the case. A lot of SIC algorithm do not use atmospheric corrections.
-Page 2, Introduction, last paragraph, line 10: “...were calculated using the existing dielectric constant model.” There is a lot of dielectric constant models that exist, please cite which one you use.
-Page 3, section 2, 2nd paragraph, line 2: “We collected the brightness temperature ...” please give some references for the SIC products level 2, and where to access the data.
-Page 3, section 2, 2nd paragraph, line 5: At how many kilometers from the coast and sea ice margins do you filter out your data?
-Page 4, section 2, 3nd paragraph, line 1: “… we simulated the BT based on the radiative transfer model (RTM)…” again, please cite the one you use. Note that several RTMs for ocean surface exist and their accuracy is decreased in cold waters, see Kilic et al., 2019.
Kilic, L., Prigent, C., Boutin, J., Meissner, T., English, S., & Yueh, S. (2019). Comparisons of ocean radiative transfer models with SMAP and AMSR2 observations. Journal of Geophysical Research: Oceans, 124(11), 7683-7699.
-Page 4 and 5: I will advise to not spend so much time to describe the ocean radiative transfer model, as you use the one from Meissner and Wentz, 2012, it has already being described. Maybe summarize this part.
Page 6, section 2, last paragraph, line 6: Please add a sentence to tell that the sensitivity to SST is larger at V-polarization. It will help for the interpretation of the results.
Page 7, section 3, 2nd paragraph, line 8: “The tie-points are usually known as static values in time and space.” Some of the SIC algorithms already use dynamic tie points in space and time see Lavergne et al., 2019.
Lavergne, T., Sørensen, A. M., Kern, S., Tonboe, R., Notz, D., Aaboe, S., ... & Pedersen, L. T. (2019). Version 2 of the EUMETSAT OSI SAF and ESA CCI sea-ice concentration climate data records. The Cryosphere, 13(1), 49-78.
Page 10, section 4, paragraph 4, line 4: Just a typo “mbT”→ “mBT”
Page 11, conclusion, last paragraph, line 6: Note that L-band is sensitive to the sea ice thickness whereas 6.9 GHz is not.
Author Response
Dear Reviewer
We would like to thank you for the thoughtful guidance. We are really grateful for the detailed comments we have received and these made a much better paper. According to the comments, we have revised our manuscript. Please see the attachment.
Sincerely,
Hyun-Cheol Kim, Ph.D
Director, Center of RS & GIS, KOPRI
